# ComFC mediates transport and handling of single-stranded DNA during natural transformation

Prashant P. Damke [1,2,7], Louisa Celma [3,7], Sumedha M. Kondekar [1,2], Anne Marie Di Guilmi[1,2], Stéphanie Marsin [3], Jordane Dépagne[4,5], Xavier Veaute [4,5], Pierre Legrand [6], Hélène Walbott [3], Julien Vercruyssen[3], Raphaël Guérois [3], Sophie Quevillon-Cheruel[3 ✉] & J. Pablo Radicella [1,2 ✉]

The ComFC protein is essential for natural transformation, a process that plays a major role in the spread of antibiotic resistance genes and virulence factors across bacteria. However, its role remains largely unknown. Here, we show that *Helicobacter pylori* ComFC is involved in DNA transport through the cell membrane, and is required for the handling of the single-stranded DNA once it is delivered into the cytoplasm. The crystal structure of ComFC includes a zinc-finger motif and a putative phosphoribosyl transferase domain, both necessary for the protein's in vivo activity. Furthermore, we show that ComFC is a membrane-associated protein with affinity for single-stranded DNA. Our results suggest that ComFC provides the link between the transport of the transforming DNA into the cytoplasm and its handling by the recombination machinery.

[1] Université Paris-Saclay, CEA, Stabilité Génétique Cellules Souches et Radiations, Institut de Biologie François Jacob, F-92260 Fontenay aux Roses, France. [2] Université de Paris-Cité, CEA, Stabilité Génétique Cellules Souches et Radiations, Institut de Biologie François Jacob, F-92260 Fontenay aux Roses, France. [3] Université Paris-Saclay, CEA, CNRS, Institute for Integrative Biology of the Cell (I2BC), F-91198 Gif-sur-Yvette, France. [4] Université Paris-Saclay, CEA, INSERM, Stabilité Génétique Cellules Souches et Radiations, Institut de Biologie François Jacob, F-92260 Fontenay aux Roses, France. [5] Université de Paris Cité, CEA, INSERM, Stabilité Génétique Cellules Souches et Radiations, Institut de Biologie François Jacob, F-92260 Fontenay aux Roses, France. [6] Synchrotron SOLEIL, L'Orme des Merisiers, F-91192 Gif-sur-Yvette, France. [7] These authors contributed equally: Prashant P. Damke, Louisa Celma. ✉email: sophie.quevillon-cheruel@i2bc.paris-saclay.fr; pablo.radicella@cea.fr

Bacterial populations display an amazing capacity to adapt to changes in their environment. In pathogens, this is reflected in the generation of variants able to colonise new hosts and in the propagation of virulence factors and the acquisition antibiotics resistance. A key mechanism in the propagation of those traits is horizontal gene transfer (HGT). There are three major mechanisms of HGT in bacteria: conjugation, phage transduction and natural transformation (NT). Although NT has been documented in at least 80 bacterial species[1], many aspects of the underlying mechanisms and players remain to be unveiled. Unlike the other pathways of HGT, NT only requires proteins coded by the recipient cell. It relies on the presence of a sophisticated apparatus capable of capturing DNA present in the environment and integrating it into the bacterial chromosome[2].

In both Gram-positive and -negative species, NT can be divided in four distinct steps[2]. It is initiated by the capture of exogenous double-stranded DNA (dsDNA) at the surface of the cell through the binding to macromolecular complexes. During the uptake step, dsDNA is imported into the periplasm (defined here as the compartments between the outer and inner membranes in Gram-negative bacteria and between the cell wall and the membrane in Gram-positive bacteria[2]). The incoming DNA is then directed to the inner cell membrane for its transport into the cytoplasm as single-stranded DNA (ssDNA). There, it is handled to the recombination machinery leading eventually to its incorporation into the chromosome, the last step of the process.

At each step of NT specialised proteins are required. Some of them are common to most species studied, but others are species-specific. While type IV pseudo-pili components have been shown to mediate the binding of the DNA to the cell surface in most competent bacterial species[3–7], this observation cannot be generalised to all competent bacteria. In *Bacillus subtilis*, wall teichoic acids, but not the pseudo-pilus, are involved in the initial binding of the DNA[8,9]. Actually, in many cases the molecules responsible for DNA capture remain to be identified. *Helicobacter pylori*, a species characterised for its high capacity for NT, does not harbour genes coding for type IV pilus or pseudo-pilus. As for the uptake step, in nearly all the naturally transformable bacteria it is mediated by a type IV pseudo-pilus[2,10], with the exception of *H. pylori* that uses a type IV secretion system to pull the DNA into the periplasm[11–13]. To complete this step, the conserved DNA receptor ComEA, a periplasmic or membrane-associated protein in Gram-negative or Gram-positive bacteria, respectively, is required. Here again, *H. pylori* constitutes an exception, since a unique protein, ComH, takes this role[14]. The transport step is carried out by the membrane channel ComEC[10,13,15–17]. ComEC is present and required for NT in all naturally transformable bacteria. Finally, the handing of the incoming DNA to the recombination machinery requires DprA[18], another transformation-specific protein with orthologues in all naturally transformable species.

Other proteins essential for NT have been identified by genetics approaches, but their role is still unknown. One of them, ComFC, is present in all naturally competent bacteria. The requirement of this protein for transformation was discovered almost 30 years ago[19,20]. The *B. subtilis* ComF locus consists of an operon harbouring three open reading frames coding for the proteins ComFA, ComFB and ComFC. All three are required for competence[20,21]. ComFA, which appears to be present in all naturally transformable Gram-positive bacteria but has not been identified in Gram-negative species, is a DNA-dependent ATPase[22–24]. A regulatory function has been proposed for ComFB[21]. In the case of ComFC, its function remains unknown despite being the only one of the three for which orthologues have been identified and described as essential for competence in all naturally transformable species studied.

Here, we characterised the ComFC from *H. pylori* and investigated its role in NT. Through the determination of its 3D structure we found that ComFC harbours phosphoribosyl transferase (PRT) and Zn-finger domains, both of which are essential for transformation. We show that in the absence of ComFC, not only the transport of the transforming DNA (tDNA) into the cytoplasm is blocked, but also, when the DNA is directly delivered to the cytoplasm, its integration into the bacterial chromosome is impaired. These phenotypes, together with the observations of a ComEC-dependent ComFC association with the inner cell membrane and its capacity to bind ssDNA suggest a model in which ComFC role provides a link between the transport and recombination steps during NT.

## Results and discussion

**ComFC participates in DNA transport through the inner membrane.** Despite the critical role of natural transformation in bacterial evolution and in the propagation of virulence and antibiotic resistances, many aspects of the transforming DNA uptake and processing remain poorly understood. A notorious example is the role of the ComFC protein. While the gene coding it was identified almost 30 years ago as essential for competence[19,20] and is conserved in all naturally transformable species studied so far, the role of the protein remains unknown. A transposon mutagenesis screen originally identified *hp1473*, a *comFC* orthologue, as a gene essential for natural transformation in *H. pylori*[25]. In the present study we undertook the characterisation of this protein and its function by a combination of approaches to (i) define the step(s) at which the protein acts during NT, (ii) determine its 3D structure and (iii) analyse its biochemical properties.

To confirm the effect of *comFC* inactivation on *H. pylori* NT, a *hp1473* null mutant was generated by insertion of a non-polar cassette and NT frequencies were determined using genomic DNA from a streptomycin resistant strain as transforming DNA (tDNA). The absence of ComFC led to an almost four log decrease in the transformation efficiency when compared to the wild-type strain (Fig. 1a). This phenotype, although less severe than that induced by inactivation of *comB, comEC* or *recA*, was similar to that observed for Δ*comH* and Δ*dprA* strains. Wild-type levels of transformation were restored by the re-expression of the *comFC* gene introduced with its own promoter at the *rdxA* locus (Fig. 1b), ruling out polar effects of the deletion.

We then sought to define at which stage(s) during the transformation process ComFC is required. Deletion of *comFC* did not affect the uptake step as illustrated by the presence of transforming DNA foci (Fig. 2a, b). Consistently with the cytoplasmic localisation of ComFC and the two-steps model for DNA uptake[13,26], the wild-type levels of DNA foci in Δ*comFC* strains indicated that ComFC is not required for the capture and uptake of the exogenous DNA. The same conclusion was reached by analysing the presence of tDNA in Δ*comFC* mutant strains of *V. cholerae* by PCR[10]. However, when the persistence of the foci was monitored, we observed that in the Δ*comFC* strain foci were detectable for longer times than in the wild type (Fig. 2b, c), similar to what was observed in a *comEC* mutant[16]. This suggested that ComFC is needed for efficient transport of the incoming DNA through the inner membrane. Consistently, when the kinetics of fluorescent DNA internalisation were followed in living bacteria[16], we observed that in the Δ*comFC* mutant, as it is the case in the Δ*comEC* strain, the transforming DNA could not be detected as entering into the cytoplasm (Fig. 2d and Supplementary Movies 1–3). These observations are very similar to those described in Δ*comEC* strains[13,16], suggesting that ComEC and ComFC act at the same NT step to mediate DNA

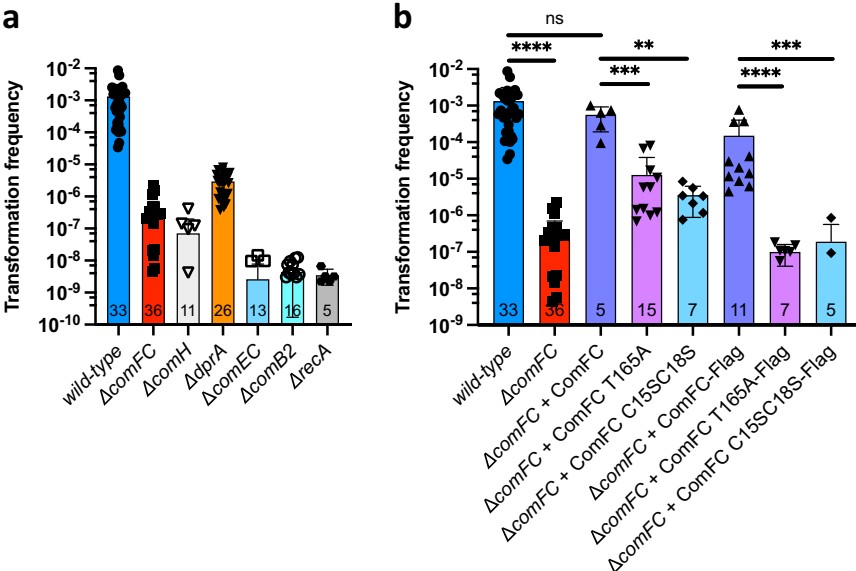

**Fig. 1 ComFC is essential for genetic transformation of *H. pylori*. a** Natural transformation frequencies for indicated *H. pylori* strains. **b** Complementation of the Δ*comFC* strain. Bars correspond to the mean and standard deviation of the number of experiments indicated within the bars. ns, not significant (*P* > 0.05); **P < 0.01, ***P < 0.001 and ****P < 0.0001. *P* values were calculated using the two-tailed Mann–Whitney *U* test on GraphPad Prism software. Source data are provided as a Source Data file.

transport across the inner membrane. Taken together, these results indicate that ComFC participates in the passage of the transforming DNA into the cytosol.

**ComFC is associated with the inner membrane**. A role of ComFC in the tDNA translocation from the periplasm to the cytoplasm supposes a connection of the protein with the membrane. To explore such a possibility, we developed an antibody against ComFC. Initially we were not able to detect the protein by immunoblot with this antibody, but by skipping the boiling step, a specific, albeit weak, signal was detectable in the wild-type strain extract (Supplementary Fig. 1). We then fractionated the extracts into soluble and membrane fractions and we observed that ComFC was associated with the membrane fraction (Fig. 3a). Further fractionation showed that ComFC co-purified with the inner membrane (Fig. 3b). To obtain a better signal, ComFC fused to a FLAG tag (ComFC-FLAG) was ectopically expressed from the *ureA* promoter. ComFC-FLAG complementation of the *comFC* deletion was less efficient, but could still support high levels of transformation (Fig. 1b). ComFC-FLAG was also present in the membrane fraction, although it could also be found in the soluble fraction probably due to its overexpression (Fig. 3b, c).

The association of ComFC with the membrane is consistent with the role of the protein facilitating the transport of the exogenous DNA through the inner cell membrane. The link to the membrane could be direct or mediated by either another protein or the incoming DNA. A candidate for coupling ComFC to the membrane in Gram-positive bacteria is ComFA. In *B. subtilis* ComFA was shown to be a membrane protein[23] and its *S. pneumoniae* orthologue interacts with ComFC[22]. While no orthologue of ComFA has been so far found in Gram-negative naturally transformable bacteria, ComEC is universally conserved in competent species, providing ComFC with a potential link to the membrane. To test this hypothesis, we analysed the localisation of ComFC-FLAG in a *comEC* mutant strain. As shown in Fig. 3c and Supplementary Fig. 2a, in the absence of ComEC, the presence of ComFC-FLAG in the membrane fraction was significantly reduced (−64%), indicating that the association of ComFC with the membrane is at least partially mediated by

ComEC. While this suggests that ComFC directly interacts with ComEC, we cannot rule out that this ComEC-dependent association with the membrane requires a yet to be identified homologue of ComFA or some other unrelated protein.

**ComFC is required for tDNA handling within the cytoplasm**. While the experiments described above demonstrated a role of ComFC in the internalisation of the transforming DNA, they do not rule out its involvement in downstream steps. In order to test this possibility, the internalisation step, impaired in the Δ*comFC* mutant, needs to be bypassed. The transforming ssDNA was therefore delivered to the cytoplasm by electroporation. While ssDNA is a poor substrate for NT[27], electroporation with a 75-mer ssDNA carrying a streptomycin resistance marker allowed transformation of mutants deficient in the uptake (Δ*comB2*) and internalisation (Δ*comEC*) steps[14]. However, after electroporation with the same ssDNA, either much fewer or no streptomycin resistant transformants were observed for mutants affecting the homologous recombination process (Δ*dprA*, Δ*recA*) (Fig. 4a). When the Δ*comFC* mutant was electroporated with the same ssDNAs, the level of streptomycin resistant recombinants was similar to that obtained with a Δ*dprA* strain. Ectopic expression of ComFC from the *rdx* promoter restored the capacity of Δ*comFC* cells to integrate the electroporated ssDNA (Fig. 4a). The reduced efficiency of a Δ*comFC* mutant in transformation by electroporation with single-stranded DNA suggests that ComFC is involved in NT steps downstream of the transport through the inner membrane.

To further explore this role of ComFC in the handling of the tDNA in the cytoplasm, we purified ComFC and analysed its capacity to bind DNA by electrophoretic mobility shift assays (EMSA). ComFC formed discrete nucleoprotein complexes with a 62-mer single-stranded DNA (ssDNA) in a concentration-dependent manner, while no binding to dsDNA was detectable (Fig. 4c). ComFC bound single-stranded oligonucleotides with relatively high affinity (half-maximal binding concentration of 300 nM). This marked preference for ssDNA is consistent with the fact that during NT the incoming DNA enters the cytoplasm as ssDNA[28].

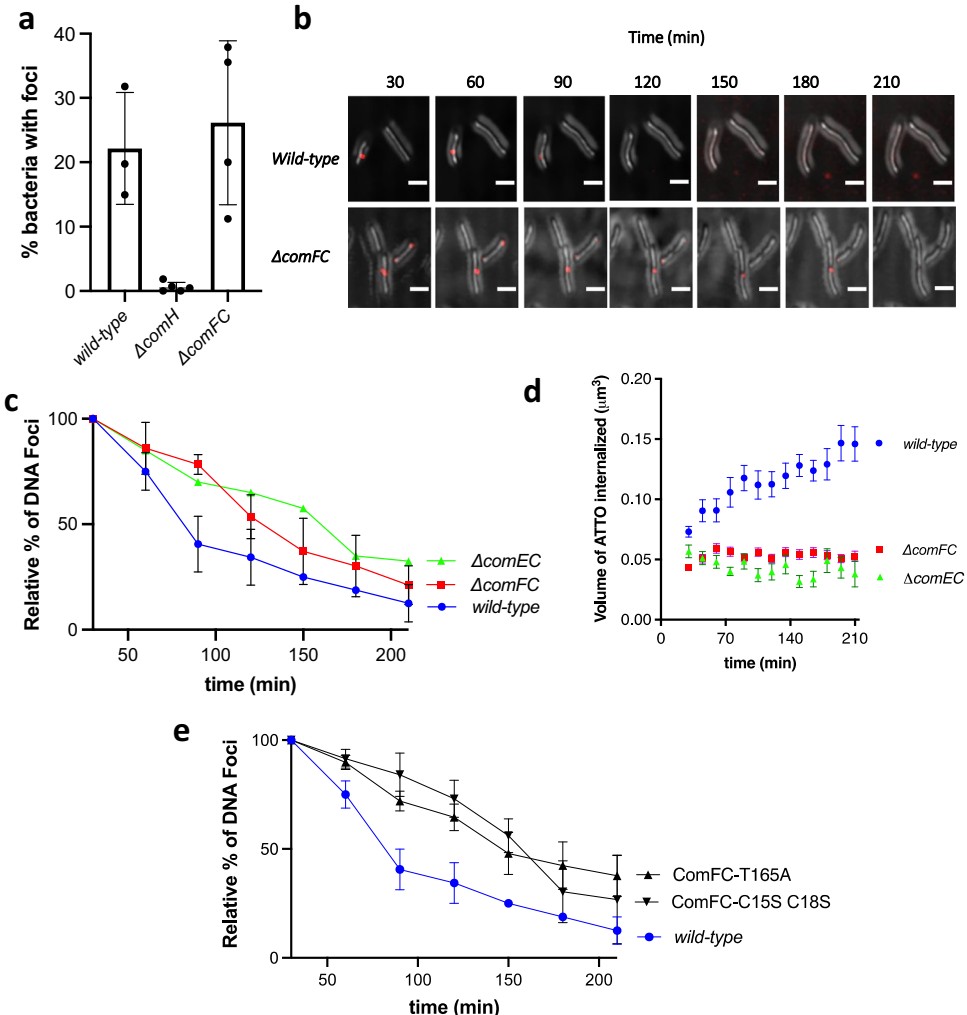

**Fig. 2 ComFC supports translocation of tDNA across the cytoplasmic membrane. a** Percentage of cells with fluorescent DNA foci for the indicated *H. pylori* strains. Bars correspond to the average and standard deviation from independent biological experiments. **b** Time course of DNA foci presence for indicated *H. pylori* strains. Z maximum projections of merged images of ATTO-550 (red channel) and differential interference contrast (DIC) are presented. Scale bars: 2.5 μm. **c** and **e** Stability of DNA foci displayed by *H. pylori* strains. Data points correspond to the mean and standard deviation from independent experiments (n = 2 for *wild-type*, n = 3 for *comFC*, n = 1 for *comEC*, n = 5 for *comFC-T165A* and n = 2 for *comFC* C15S C18S. **d** Internalisation kinetics of fluorescent DNA in indicated *H. pylori* strains. GFP expressing bacteria displaying fluorescent DNA foci were followed for 3 h by confocal microscopy in live conditions (Supplementary Movies 1–3). The mean ± SEM for volumes of DNA internalised were measured by 3D-analysis of individual bacterial cells for wild-type (n = 26), Δ*comEC* (n = 25), Δ*comFC* (n = 148) strains. At least two independent experiments were performed for each strain except for Δ*comEC*. *P*-values calculated using Kruskal–Wallis statistics indicate that Δ*comEC* (p < 0.0001) and Δ*comFC* (p = 0.0003) curves are significantly different from the wild-type curve. Source data are provided as a Source Data file.

The failure to bypass the transformation defect of the Δ*comFC* mutant by electroporation with ssDNA, together with the capacity of ComFC to bind ssDNA, indicated that ComFC is likely to be implicated in the steps leading to the formation of the recombination substrate within the cytoplasm. ComFC could, together with DprA and RecA[29], participate in the protection of the incoming DNA from degradation. We therefore tested the capacity of the purified ComFC to protect ssDNA from mung bean non-specific nuclease. Albeit with less efficiency than the tested SSB, ComFC blocked the nuclease action on a ssDNA 78-mer oligonucleotide (Fig. 4d). Interestingly, ComFC from *Campilobacter jejuni* and ComFC from *S. pneumoniae*, both required for natural transformation[22,30], interact with DprA[22,31]. Since DprA plays a critical role in the loading of the recombinase to the transforming DNA[18], it is tempting to speculate that ComFC not only binds the tDNA emerging from ComEC into the cytoplasm to protect it against degradation but also targets it to DprA to allow further processing by the recombination machinery.

**ComFC harbours zinc-finger and PRT domains**. Despite its conservation amongst naturally competent bacteria (Supplementary Fig. 3), no structural data on ComFC proteins is available. The determination of its 3D structure has been elusive. Indeed, despite many efforts to improve the purification protocol and storage buffer, in our hands the purified ComFC protein from *H. pylori* turned out to be unstable at concentrations higher than 1 mg/ml, leading to its precipitation and, therefore, preventing from reaching a concentration suitable for nucleation in the crystallisation tests. Thus, in order to help crystallisation, we generated a gene fusion between the full-length *comFC* gene and an artificial αRep binder coding sequence selected from a highly diverse library of artificial repeat proteins based on thermostable HEAT-like repeats[32,33]. Since structural homology predictions using HHpred (toolkit.tuebingen.mpg.de/hhpred/)[34] suggested that the C-terminal domain of ComFC (residues 53 to 188) harbours a putative nucleoside binding site characteristic of phosphoribosyl transferases (PRTases) belonging to the PRT

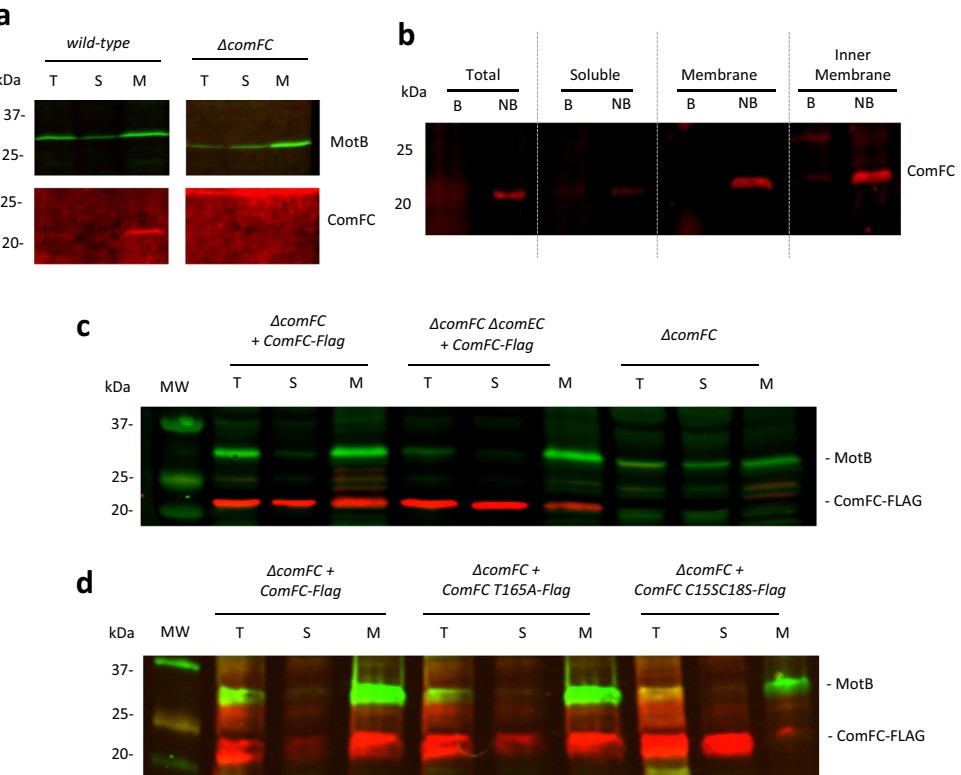

**Fig. 3 ComFC is a membrane-associated protein. a** and **b** Western blot analysis of subcellular fractions. Localisation of ComFC in wild-type *H. pylori* strain using an anti-ComFC antibody. Blots are representative of two fractionation experiments. **c** and **d** Localisation of overexpressed ComFC-FLAG wild-type and mutant proteins using an anti-FLAG antibody. Blots are representative of 7 and 9 experiments for panels **c** and **d**, respectively. The inner membrane protein MotB was used as marker for the fractionation experiments. T: total extract, S: soluble fraction, M: membrane. B: boiled samples, NB: not boiled samples. Source data are provided as a Source Data file.

family (PurF, PDB number 6CZF-A, probability 99.15%), crystals were grown in the presence of either AMP or 5-phospho-α-D-ribosyl 1-pyrophosphate (PRPP). In the presence of AMP the resulting crystals were of poor quality (about 8 Å resolution). Diffracting crystals were however obtained with the purified fusion protein in the presence of PRPP and the 3D structure of full-length ComFC in complex with PRPP was solved at 2.5 Å resolution (Table 1, Materials and Methods)[35]. The fusion (αRep: residues 1–229, linker: residues 230–236, ComFC: residues 237–427) is present in four copies in the asymmetric unit, organised in two domain-swapped dimers: the αRep of one fusion covers the ComFC of the other one (Fig. 5a and Supplementary Fig. 4). The αRep-ComFC fusions interact within the two dimers with interface areas of 2285 Å$^2$ and 2192 Å$^2$, respectively (PDBePISA server[36]). In comparison an interface of only 526 Å$^2$ connects both dimers together, demonstrating that the tetrameric form is probably due to the crystal packing (Supplementary Fig. 4a). The 3D structure obtained confirmed the presence of two predicted distinct domains in ComFC, a N-terminal Zn-finger-containing domain and a C-terminal PRT domain, which are described below.

The presence of the αRep impeded to conclude from the crystal structure on the possibility of ComFC adopting higher order quaternary structures. Most PRT proteins form dimers[37]. This, together with the dimers described for the *S. pneumoniae* ComFC protein[22], prompted us to explore by bacterial two hybrid assays (BacTH) whether the *H. pylori* orthologue could also interact with itself. This was indeed the case (Fig. 5b), indicating that, as in the case of *S. pneumoniae* ComFC, ComFC from *H. pylori* could form dimers.

**The zinc-finger domain is required for ComFC function**. The small NTD of ComFC (residues M1–D21), which is part of a larger domain corresponding to the additional and variable "hood" domain of the PRTase family[37], is a 4-Cys Zn-finger (in grey and pink in Fig. 5c, d). The Zn-finger is connected by a five amino acid linker to the rest of the hood domain (residues L22–T54), structured into a small sheet of two β strands followed by a kinked α helix. A Zn$^{2+}$ ion is liganded into the protein via the four cysteine residues (C3, C6, C15 and C18 in pink in Fig. 5c, d) which are highly conserved within the ComFC family (Supplementary Fig. 3).

To assess the role of the zinc-finger, we expressed from either the *rdxA* or the *ureA* loci a ComFC version in which cysteines 15 and 18 were replaced by serine (ComFC-C15S,C18S), and tested its capacity to complement the transformation phenotype of a *ΔcomFC* strain. Unlike the wild-type ComFC, the mutant protein could not restore natural transformation to that of the equivalent construct expressing the wild-type protein (Fig. 1b). To further explore the role of this domain, we analysed the persistence of DNA foci in the strain expressing ComFC-C15S C18S-Flag. As observed in the case of the *ΔcomFC* strain, the DNA foci formed in the strain expressing ComFC with the disrupted Zn-finger domain displayed a much longer half-life than in the wild type strain (Fig. 2e). Surprisingly, mutation of the cysteine residues did not affect the integration into the bacterial chromosome of a ssDNA delivered by electroporation into the cytosol (Fig. 4b). These results suggest that the Zn finger motif is necessary for the efficient internalisation of the tDNA into the cytoplasm, but dispensable for ComFC role in the downstream steps. Zinc-finger domains are most often found in proteins known to bind DNA or

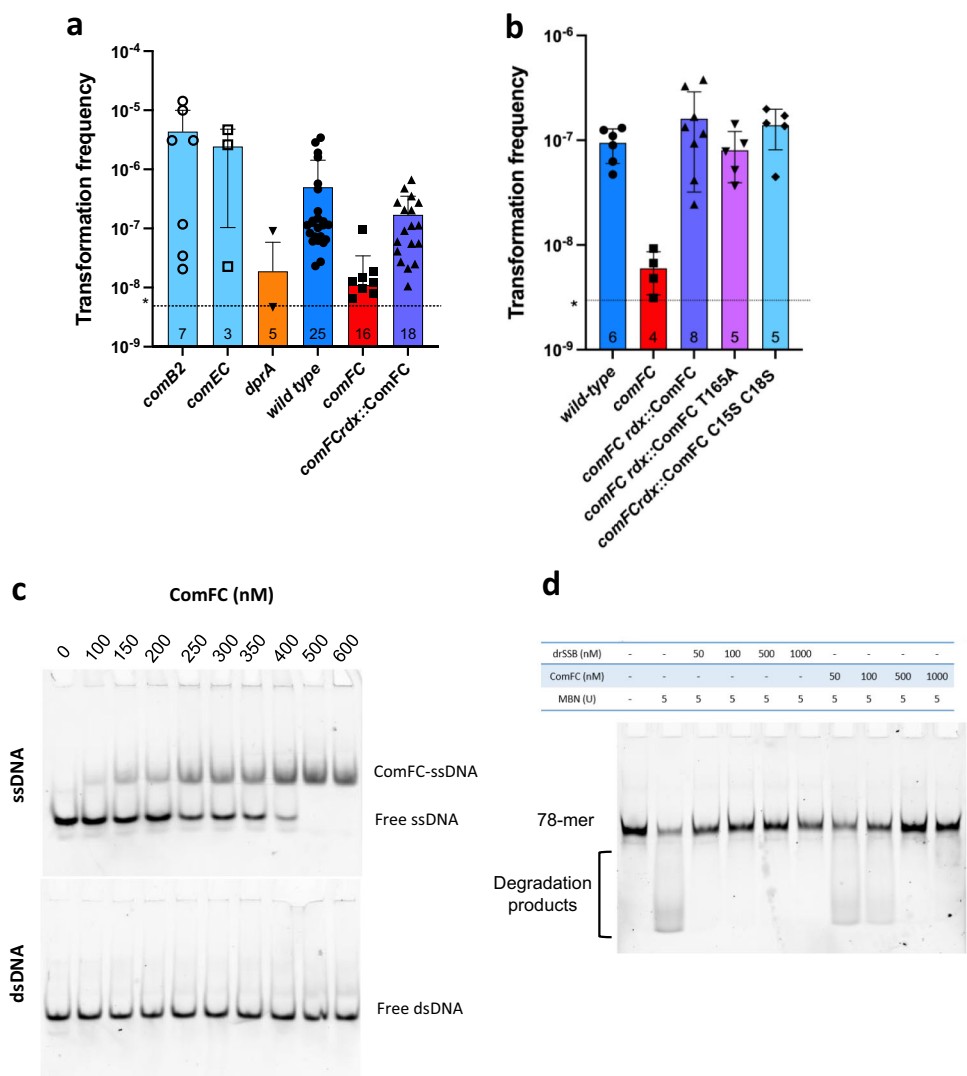

**Fig. 4 ComFC binds single-stranded DNA and promotes its chromosomal integration. a** and **b** Transformation frequencies after electroporation with a single-stranded DNA (75-mer) coding for streptomycin resistance as donor DNA. Bars correspond to the average and standard deviation from the number of experiments indicated within the bars. * Assay detection limit as determined considering one colony obtained in a *recA* mutant. **c** Selective affinity of ComFC for single-stranded DNA. Indicated concentrations of His$_6$-ComFC were incubated with a Cy5-labelled single- or double-stranded DNA oligonucleotide. The nucleoprotein complexes were resolved by native-PAGE. **d** Protection against nuclease activity. 78 nt long fluorescent oligonucleotide was incubated 50 min with 5 units of Mung Bean Nuclease (MBN) after 20 min of preincubation with drSSB or ComFC at the indicated concentrations. Products were analysed on a native polyacrylamide gel. Blots are representative of two experiments. Source data are provided as a Source Data file.

RNA[38]. In particular, 4-Cys zinc-fingers are present in ribosomal proteins or in enzymes involved in DNA replication, recombination and transcription[39,40]. Unfortunately, attempts to purify ComFC versions either mutated in cysteines 15 and 18 or deleted of the zinc-finger domain were unsuccessful, preventing further exploration of its function at the biochemical level. Remarkably, using the dimeric fusion protein, in which the αRep domain totally masks the zinc-finger domain of ComFC (Fig. 5a and Supplementary Fig. 4a), no interaction with DNA was observed. The αRep domain could therefore prevent access to the DNA-binding site of ComFC. There are, however, several examples of zinc-finger domains that do not participate in nucleic acid binding, but are involved in protein-protein interactions[41,42]. Mutation of the two cysteines in the zinc-finger domain of ComFC significantly reduced the association of the protein with the membrane fraction (Fig. 3d and Supplementary Fig. 2b), suggesting that the Zn-finger domain is required for the interaction of ComFC with either ComEC or the protein that

mediates this interaction. An interesting case is that of RadA, a DNA helicase harbouring a 4-Cys domain and implicated in NT of Gram-positive bacteria. When mutated in its 4-Cys domain RadA is still able to bind DNA and to carry out its ATPase and helicase activities, but it cannot interact with RecA, thus limiting its D-loop unwinding capacity[43].

The closest structural homologue of the ComFC zinc-finger is the zinc-finger domain of RecR, a recombination protein (RMSD of 0.8 Å for 22 aligned residues, PDB number 4O6O[44] or PDB number 5Z2V[45]). While the role of the RecR zinc-finger remains to be determined[46,47], it has been suggested that it has a structural role in protein folding[47]. Supporting this possibility, as in our case, the authors were unable to produce soluble forms of *E. coli* or *H. pylori* RecR mutated in its zinc-finger cysteines. While our data supports a role of this domain in the interaction of ComFC with other NT partners, we cannot completely rule out that ComFC Zn-finger is involved in the binding of the transforming DNA.

**Table 1 Data collection and structure refinement statistics.**

| Data collection | PRPP-bound B2-HpComF ¤ |
|---|---|
| Space group | $P1$ |
| Unit cell parameters | $a = 58$Å $b = 88$Å $c = 123$Å |
| | $\alpha = 80°$ $\beta = 76°$ $\gamma = 76°$ |
| Wavelength (Å) | 0.979 |
| Resolution range (Å)† | 46.5–2.5 (2.6–2.5) |
| *Before STARANISO* | |
| Measured/Unique reflections † | 895493/77488 (57841/5398) |
| Completeness (%)† | 98.4 (92.6) |
| Anomalous completeness (%)† | 95.0 (84.8) |
| $I/\sigma(I)$† | 15.7 (0.6) |
| *After STARANISO* | |
| Measured/Unique reflections † | 752919/63154 (44297/3158) |
| Completeness (%)† | 93.5 (94.2) |
| Anomalous completeness (%)† | 90.6 (92.7) |
| $I/\sigma(I)$† | 19.2 (1.6) |
| Redundancy † | 11.9 (14.0) |
| Anomalous redundancy † | 6.1 (7.1) |
| $CC_{1/2}$ † | 0.999 (0.387) |
| $CC_{ano}$ † | 0.715 (0.403) |
| $|DANO|/\sigma(DANO)$ | 1.404 (0.814) |
| $R_{merge}$ (%)† | 8.8 (140.1) |
| $R_{pim}$ (%)† | 2.7 (38.3) |
| **SAD phasing** | |
| Number of sites | 12 |
| Overall FOM | 0.39 |
| **Refinement** | |
| Resolution range (Å)† | 46.5–2.5 (2.6–2.5) |
| Number of work/test reflections | 63118/3156 |
| $R/R_{free}$ (%)† | 22.1/24.5 (25.6/27.2) |
| **Geometry statistics** | |
| Number of atoms | |
| Protein | 12672 |
| Ligand/ion | 184 |
| Water | 282 |
| r.m.s. deviations from ideal values | |
| Bond lengths (Å) | 0.007 |
| Bond angles (°) | 0.87 |
| Average B-factor (Å²) | |
| Overall | 69.74 |
| Protein | 69.92 |
| Ligand/ion | 79.81 |
| Water | 54.88 |
| **Ramachandran plot** | |
| Favoured (%) | 97 |
| Allowed (%) | 3 |
| Outliers (%) | 0 |

†Values in parentheses refer to the highest resolution shell.
¤ Four merged diffraction datasets collected from one crystal, which diffracted anisotropically to 2.8 Å along *0.864 a\* − 0.025 b\* + 0.503 c\**, 2.7 Å along *0.168 a\* + 0.983 b\* + 0.067 c\** and 2.3 Å along *−0.018 a\* + 0.097 b\* + 0.995 c\**.

**The PRT domain in necessary for ComF function.** The ComFC CTD (residues L55–D190, the terminal E191 is not defined in the electron density) shares the common core of the amidophosphoribosyl transferase type 1 fold (RMSD between 2.6 and 3 Å for 100 to 130 aligned residues, PDB number 5ZGO[48] as an example, according to the DALI server[49]).

The central parallel β sheet characteristic of the PRTase core domains is present in ComFC (β strands [183]AIA[185], [153]YFLLD[157], [85]LYGIA[89] and [113]LKP[115], in cyan in Fig. 5c), extended by the two β strands of the NTD ([27]KVRVL[31] and [34]VSVYS[38], in grey in Fig. 5c). The three Mg•PRPP-binding loops of the family are present, providing a large hydrogen bonds network with the PRPP (in dark blue in Fig. 5c, e, and in Supplementary Fig. 4b). An electron density that can correspond to a $Mg^{2+}$ ion is present close to the PRPP. The "PRPP loop" carries the canonical [157]DDIITTGTTL[166] active site signature allowing the binding of the ribose-5-phosphate group of the PRPP (Supplementary Figs. 3 and 4b). The most variable "PPi loop" (A[89] to H[100]) allowing the binding of the PPi group of the PRPP is slightly longer than the standard four amino acids loops. The "flexible loop" (L[120] to T[144]) closes the pocket of the binding site occupied in our structure by the PRPP (red sticks in Fig. 5c, e). The presence of all three loops (Supplementary Fig. 3) is considered the signature of the PRT family[37].

The PRT domain, present in a large variety of proteins, is known to bind small molecules such as nucleotides, glutamine or NMPs in vitro[37,50]. Since no structural variation of the binding site is observed between the proteins harbouring it (Supplementary Fig. 4c), its 3D structure is not sufficient on its own to predict the cognate ligand in vivo. We performed differential scanning fluorimetry/thermal shift assays to detect interactions of purified *H. pylori* His$_6$-ComFC with various potential ligands. Figure 6a shows that the wild-type protein exhibited a Tm of around 46 °C. In the presence of AMP, the fluorescence maxima observed for the wild-type protein was shifted by +9 °C, suggesting the stabilisation of the protein through binding of the nucleotide. Albeit to a lesser extent, ADP addition also resulted in an increase in melting temperature (Supplementary Table 1). No effect was observed with the triphosphate nucleotides. To confirm that the nucleotide binding was through the PRPP-binding domain, we purified a mutant version of the protein where threonine in position 165 within the conserved [155]LLDDIITTGTTL[166] motif was replaced by an alanine. ComFC T165A had a melting temperature close to that of the wild-type, indicating that the amino acid replacement did not significantly affect the structure of the protein. However, addition of the nucleotides had a very modest effect on the thermal stability of the protein (Fig. 6b and Supplementary Table 1), confirming the role of the conserved threonine in ligand binding.

To assess the relevance of ComFC nucleotide binding capacity in the function of the protein during NT, we expressed in a Δ*comFC* strain either ComFC T165A from the *rdxA* locus using its own promoter or from the *ureA* locus. While expression of ComFC T165A restored to a certain extent the transformation capacity of the Δ*comFC* strain, the recombinant frequency was > 40-fold lower than that obtained with the wild-type protein expressed in the same conditions (Fig. 1b). Furthermore, expression of the mutant protein in a Δ*comFC* background resulted in a longer half-life of the periplasmic DNA foci (Fig. 2e). This latter observation indicates that the integrity of the PRPP binding site is required for ComFC role in the translocation of the tDNA from the periplasm to the cytosol. To test whether the PRPP binding domain is required for ComFC role in the handling of the single-stranded tDNA within the cytoplasm, we delivered the transforming ssDNA by electroporation to Δ*comFC* bacteria expressing ComFC-FLAG T165A. The yield of recombinant clones was equivalent to that obtained in the strain expressing the wild-type ComFC (Fig. 4b), indicating that while the presence of ComFC is required for allowing the integration of the ssDNA into the chromosome, its capacity to bind nucleotides is dispensable once the ssDNA is present in the cytoplasm.

The fact that the protein belongs to the PRT family is surprising[37]. The majority of PRT proteins are enzymes that catalyse the displacement of pyrophosphate from PRPP by a nitrogen-containing nucleophile[37]. While there are other

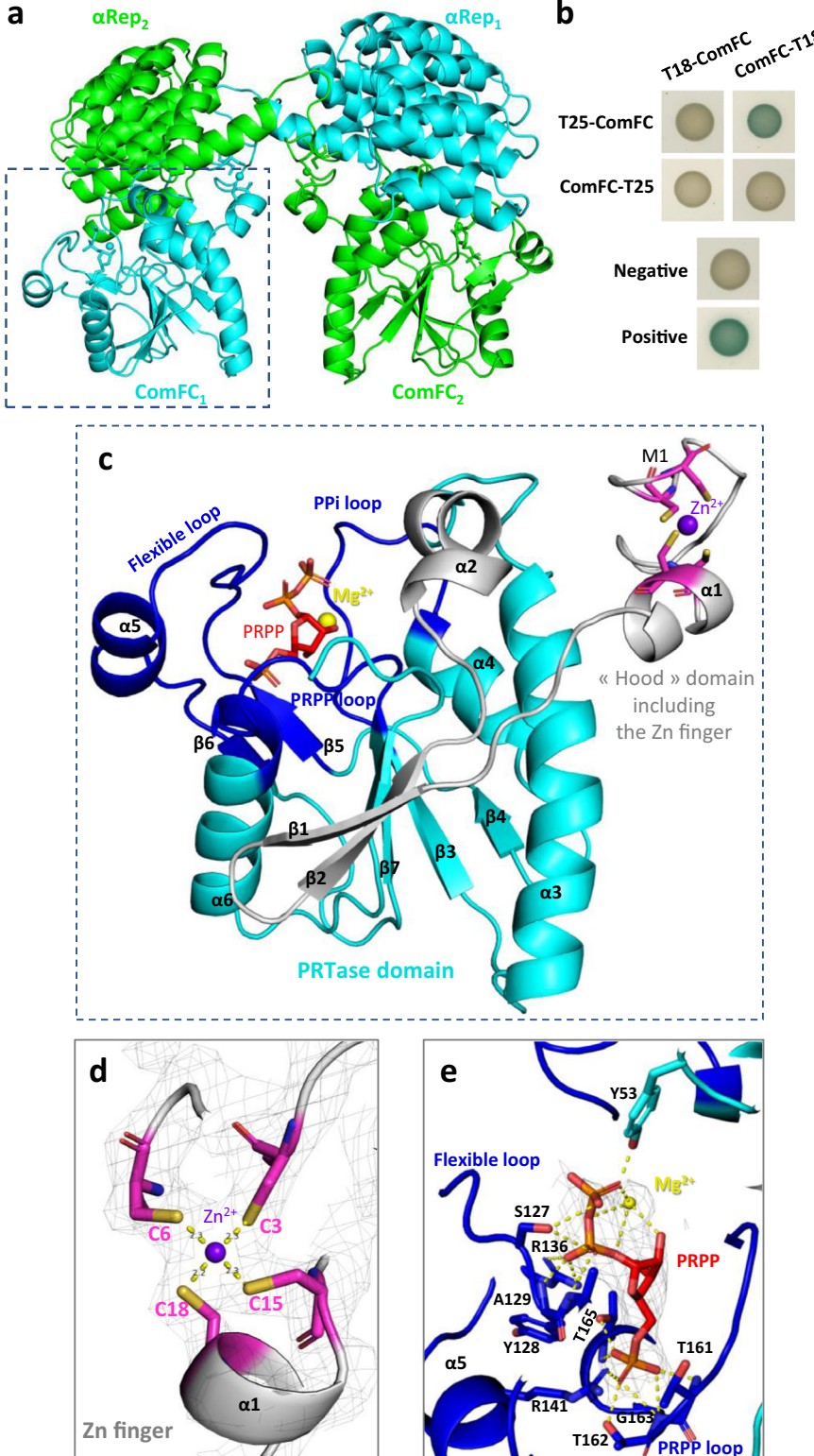

**Fig. 5 *H. pylori* ComFC harbours a PRT and Zn-finger domains. a** Crystal structure of the αRep-ComFC domain-swapped dimer. The two protein fusions are in green and blue. The αRep was evolved against ComFC to develop a specific interaction surface. In the crystal packing, each αRep returned to the ComFC of another fusion, allowing the crystallisation of an artificial dimer. The PRPP co-crystallised with the protein is represented by sticks, and the $Zn^{2+}$ ion is schematized by a sphere. **b** Bacterial two-hybrid assay of *H. pylori* ComFC. Representative images of reporter strains grown on plates supplemented with IPTG and X-Gal are shown. **c** ComFC structure. The three loops characteristic of the PRTase fold are in dark blue and the "hood" domain is in cyan. The PRPP is in red sticks and the $Zn^{2+}$ and $Mg^{2+}$ ions are represented by purple and yellow spheres, respectively. **d** $2F_o-F_c$ electron density map (grey mesh, contoured at 1.5 sigma) of the 4 cysteines, coloured as in **c**, forming 4 coordination bonds (yellow dotted lines) with the $Zn^{2+}$ ion. **e** $2F_o-F_c$ electron density omit-map (grey mesh, contoured at 1.5 sigma) of the PRPP and the $Mg^{2+}$ ion in interaction with ComFC (coloured as in **c**). The shared hydrogen bonds between ComFC amino acids (in dark blue sticks) and PRPP or $Mg^{2+}$ are shown as yellow dotted lines.

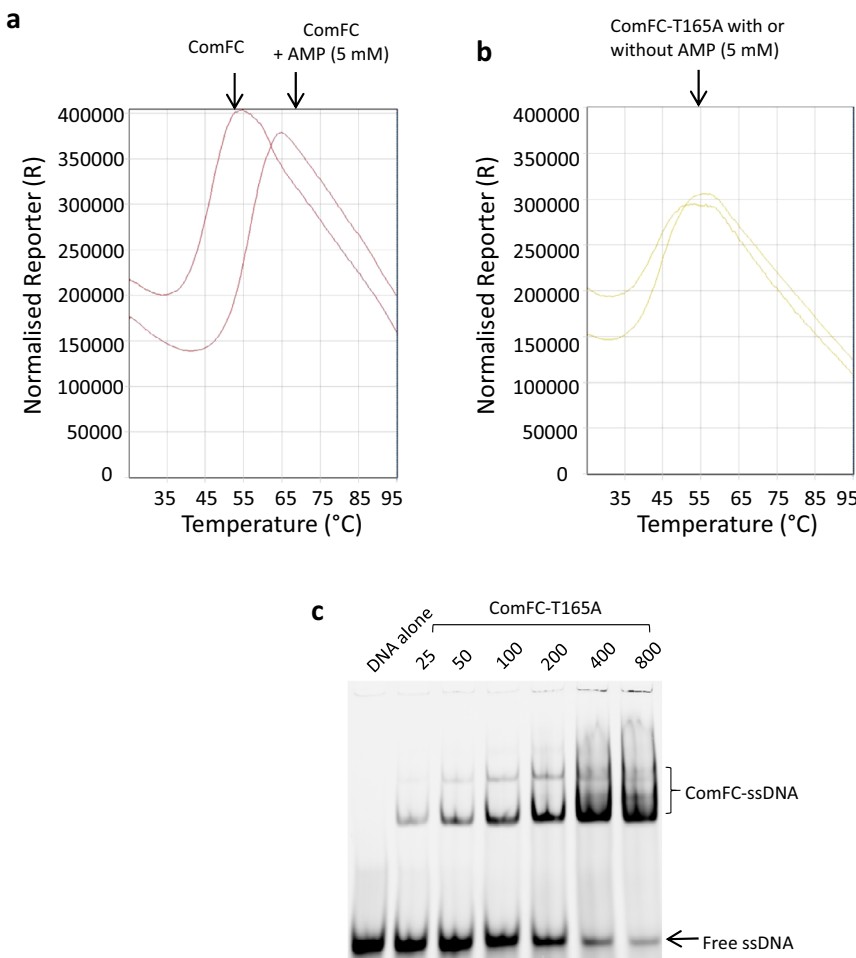

**Fig. 6 ComFC binds nucleotides through its PRTase motif.** Thermal denaturation curves displaying melting temperature of **a** ComFC and, **b** ComFC-T165A with or without Adenosine monophosphate. **c** Single-stranded DNA binding by ComFC-T165A. Blot is representative of three independent experiments.

PRTases, those belonging to the PRT family are involved in nucleotide synthesis and salvage pathways. Our results (Supplementary Table 1) show that ComFC PRT domain is capable of binding not only PRPP but also monophosphorylated nucleotides and, albeit with less affinity, nucleotide di-phosphates. The physiological ligand remains, however, to be determined, since the "PRPP loop" of the PRTase family is structurally highly conserved and does not exhibit significant ligand-dependent variation (Supplementary Fig. 4c).

In a few PRT proteins the PRTase capacity to bind PRPP or the nucleotide substrate has been co-opted for regulatory functions as described for two *Bacillus subtilis* regulators of gene expression. PurR binds to DNA operator sequences to repress the expression of purine genes. Binding of PRPP lowers its affinity for the DNA, triggering expression[51]. PyrR binds regulatory regions of pyrimidine genes transcripts attenuating their expression. Its affinity for the mRNA is regulated by UMP[52]. We thus asked if the T165A mutation affected ComFC affinity for ssDNA. Even though this substitution abolished the interaction with nucleotides (Supplementary Table 1) it did not significantly affect the binding of ssDNA (Fig. 6c and Supplementary Fig. 5). We then asked if the integrity of the PRPP binding site is required for ComFC localisation to the membrane. As shown in Fig. 3d and Supplementary Fig. 2b, the T165A mutation did not affect the targeting of the protein to the membrane.

The experiments presented here do not allow to conclude on whether ComFC PRT domain provides a PRTase activity or a regulatory function. An intriguing hypothesis is that the deoxyribomononucleotides released by the degradation of the non-transforming strand[53] might regulate ComFC capacity to bind DNA. Although ComFC T165A is not affected in its DNA affinity (Fig. 6c), it is possible that binding to the wild-type protein of a so far unidentified nucleotide results in a reduced affinity for the transforming DNA. It is worth noting that the T165A mutant, while completely impaired in nucleotide binding (Fig. 6b and Supplementary Table 1), can still partially rescue the transformation phenotype of a Δ*comFC* mutant (Fig. 1b), suggesting that nucleotide binding to ComFC could provide a fine-tuning mechanism of the transformation process. Such a scenario would be consistent with the recently proposed hypothesis that ComFC provides a link between transformation and metabolism[54].

## Conclusion

In this study, using *H. pylori* as a model, we sought to unveil the role of ComFC, one of the most conserved proteins involved in horizontal gene transfer through NT. Despite the discovery of its essentiality for competence over 30 years ago, the understanding of where and how this protein participates in NT remained elusive. We showed here that ComFC is required for at least two different steps in NT. First, ComFC facilitates the transport of the tDNA through the cell membrane. Consistent with this finding we found that the protein localises to the inner membrane.

Secondly, ComFC, which we show has affinity for ssDNA, is involved in the handling of the DNA within the cytosol. We therefore propose that ComFC provides a link between these two distinct steps during NT. Our structural studies demonstrated that ComFC is composed of two conserved domains, both essential for its in vivo activity: a 4-Cys zinc-finger domain and a PRPP-binding domain. While several details of ComFC mechanism of action remain to be elucidated, the data presented here shed light on the role of this protein critical for NT in all naturally competent bacteria.

## Methods

**H. pylori cultures**. H. pylori strains are listed in Supplementary Table 2. Cultures were grown under microaerophilic conditions (5% $O_2$, 10% $CO_2$, using the MAC-MIC system from AES Chemunex) at 37 °C. Blood agar base medium (BAB) supplemented with 10% defibrillated horse blood (AES) was used for plate cultures. Liquid cultures were grown in brain heart infusion media (BHI) supplemented with 10% defibrillated and de-complemented foetal bovine serum (Invitrogen, Carlsbad, CA, USA) with constant shaking (180 rpm). An antibiotic mix containing polymyxin B (0.155 mg/ml), vancomycin (6.25 mg/ml), trimethoprim (3.125 mg/ml), and amphotericin B (1.25 mg/ml) was added to both plate and liquid cultures. Additional antibiotics were added as required: kanamycin (20 μg/ml), apramycin (12.5 μg/ml), and chloramphenicol (8 μg/ml) streptomycin (10 μg/ml).

**Construction of gene variants in H. pylori**. All oligonucleotides and plasmids used in this work are listed in Supplementary Tables 3 and 4. H. pylori 26695 gene sequences were obtained from the annotated complete genome sequence of 26695 deposited at http://genolist.pasteur.fr/PyloriGene/. Gene/locus specific primers (listed in Supplementary Table 3) were used to amplify the region of interest by PCR, and fragments were joined together by either classical restriction-ligation method or using sequence- and ligation-independent cloning (SLIC). Tags and mutations in the genes were introduced using SLIC and site directed mutagenesis, respectively. All the plasmids generated (listed in Supplementary Table 4) were verified by sequencing. The knock-out /knock-in cassettes were introduced into H. pylori by natural transformation. Their correct integration in the genome was confirmed by PCR using locus and gene specific primers. Verified strains (Supplementary Table 2) were stored at −80 °C in BHI media supplemented with 12.5% glycerol. The details for the different constructions generated in this study are given below.

**Construction of hp1473 null mutants in H. pylori**. To generate hp1473 locus disrupted by a non-polar chloramphenicol cassette (hp1473::Cm), the hp1473 locus was amplified using primers hp1473F and hp1473R and ligated into pjET1.2 vector to generate pJET1.2-hp1473. PCR fragments generated by amplification of this plasmid (using primers 1473 inverse F and 1473 inverse R), and non-polar chloramphenicol (using primers KpnI -Cm-for and BamHI-Cm-rev) resistance cassette were digested using KpnI and BamHI and then ligated to generate the knockout cassette p978 (pJet1.2-hp1473::Cm).

To generate hp1473 locus disrupted by a non-polar apramycin cassette (hp1473::apramycin), hp1473 locus was amplified using primers Op853 and Op854 and ligated to pjET1.2 vector amplified using Op855-Op856 to generate pJet1.2-hp1473. The PCR fragments generated by amplification of pJET1.2-hp1473 (using primers Op859 and Op860) and a non-polar apramycin resistance cassette (using primers Op857 and Op858) were ligated using SLIC to generate p1699.

**Ectopic expression of hp1473 variants**. The hp1473 locus (+152 bp upstream sequence) was amplified using Op5 and Op6 and inserted in plasmid p1175 harbouring a rdxA::km cassette (amplified using Op3 and Op4) to generate plasmid p1176. T165A and C15SC18S mutations in the hp1473 coding region were introduced by PCR using mutagenic primers (Op13 + Op14 and Op302 + Op303 respectively). The native hp1473 locus was disrupted using p978. For expression using the urea promoter hp1473 loci, wild-type or containing T165A or C15SC18S point mutations, were amplified using Op611 and Op612 (containing the sequence for Flag tag) and introduced using SLIC into p1088 (amplified using Op613 and Op614) containing the Promoter-UreA-Cm cassette to generate p1672, p1674 and p1676 respectively. These plasmids were used to transform H. pylori. The native hp1473 locus was disrupted using p1699.

**Determination of transformation frequencies**. Natural transformation frequencies were determined as described[55]. Briefly, total chromosomal DNA (200 ng) from a streptomycin resistant, but otherwise isogenic strain, was incubated overnight with exponentially growing H. pylori cells (optical density of 4.0 at 600 nm), on solid medium. Next day, serial dilutions of H. pylori were spread on plates with and without 10 μg/ml streptomycin. Transformation frequencies after electroporation were determined as described[14]. Briefly, electro-competent cells were prepared by treating H. pylori cells (optical density of 10 OD/ml at 600 nm) with ice-cold Glycerol 15 % + Sucrose 9 %. 50 μl of electro-competent cells mixed with 1 μg of 75-mer-ssDNA (Supplementary Table 3) carrying A128G mutation in the hp1197 gene were electroporated at 2.5 kV cm$^{-1}$ and 25 μF. The cells were mixed with 100 μl BHI, and 50 μl were spotted on BAB plates. Next day, serial dilutions of H. pylori cells were plated on plates with or without streptomycin (10 μg/ml). The transformation frequencies were calculated as the number of streptomycin resistance colonies per recipient colony-forming unit. P values were calculated using the Mann–Whitney U test on GraphPad Prism software.

**Fluorescence microscopy experiments**. Microscopy experiments were performed as described earlier[14,16]. Fluorescent dsDNA (408 bp) was prepared by amplification of hp1197 locus from 26695 gDNA (100 ng) using primers 1197-5' and 1197-3' (0.5 μM each), 250 μM of dNTP mix, 5 Units of ExTaq enzyme (Takara) supplemented with 10 μM of ATTO-550-aminoallyl-dUTP (Jena bioscience). PCR elongation was performed at 72 °C (2 min per kb) and the amplified products were purified with the Illustra GFX purification kit (GE Healthcare Little Chalfont, UK).

Exponentially growing H. pylori cells were incubated with fluorescent DNA (200 ng) for 7 min at 37 °C, the unbound DNA was washed and the bacteria were re-suspended in BHI, covered with low melting agarose (1.4%) supplemented with 10 % foetal bovine serum and observed under live conditions (gas mixture (10% CO2, 3% $O_2$), humidity (90%)) at 37 °C for 3 h. Alternatively, the bacteria were fixed with 4% formaldehyde (90 mins at 4 °C) followed by quenching with 100 mM Glycine. All the images were captured with 60X objective using inverted Nikon A1R confocal laser scanning microscope system. The images were processed and analysed using NIS-element software (Nikon Corp., Tokyo, Japan) and ImageJ software. The percentage of bacteria with DNA foci was calculated as the number of bacteria with DNA foci over total number of bacteria counted in at least two independent biological replicates. To monitor the time dependent stability DNA foci, the total number of bacteria with DNA foci at t = 15 mins were considered as 100%. The volumes of internalised DNA in GFP expressing bacteria were estimated by 3-D image analysis performed using Volocity software (Perkin Elmer, Waltham, USA).

**Subcellular fractionation**. Subcellular fractions of exponentially growing H. pylori were collected by differential centrifugation and detergent mediated solubilisation as described earlier[14]. Briefly, 100 ml cell pellet was re-suspended in buffer A (10 mM Tris-HCl, pH 7.5, 1 mM DTT, 1X protease inhibitor cocktail) followed by lysis by sonication. Total extracts were centrifuged 14,000 rpm for 15 min. The supernatant containing the soluble fraction was collected after ultracentrifugation at 45,000 rpm for 45 min of the total extract. The pellet containing the membrane fractions was re-suspended in buffer B (10 mM Tris-HCl, pH 7.5, 1 mM DTT, 1X protease inhibitor cocktail, 1% N-Lauroylsarcosine). The supernatant containing the inner membrane fractions were collected after ultracentrifugation at 45,000 rpm for 45 min. The presence of the proteins in the various fractions was monitored by immunoblotting.

**Western blots**. Protein samples were resolved in a 15% SDS-PAG and transferred to a nitrocellulose membrane. The membrane was blocked with 2% BSA in PBST (1X PBS + 0.03% Tween 20). Blots were probed with either a mouse monoclonal anti-Flag antibody (1: 5000 dilution, Sigma Aldrich), rabbit anti-MotB antibody (1: 2500 dilution) (kind gift from Dr. Ivo Boneca, Pasteur Institute) or rabbit anti-HpComFC antibody from our laboratory collection. The blots were then probed with Advansta fluorescently labelled secondary antibodies IR700 or IR800. The imaging was done using Odissey Clx imaging system. The blots were quantified using the Image Studio software. The ratios ComFC/MotB were normalised based on the membrane enrichment as monitored by MotB presence. Values for the wild-type were set to 100%.

**E. coli cultures**. Escherichia coli strains used for cloning, protein overexpression and purification were cultured in Luria–Bertani (LB) broth, or LB agar plates supplemented with the required antibiotics [ampicillin (100 μg/ml), kanamycin (50 μg/ml), apramycin (50 μg/ml), or chloramphenicol (34 μg/ml)].

**Protein samples preparation**. Cloning of comFC (hp1473) coding region was performed using genomic DNA from H. pylori strain 26695 as template for PCR. Six histidine codons were added at the 5' end during the PCR process. The amplified fragment was inserted into the NdeI-XhoI sites of pET21a vector (Novagen). Site directed mutagenesis was performed using the resulting pET21:ComFC-6His plasmid as a template and non-overlapping oligonucleotides phosphorylated in 5' (Eurofins), to construct the HpComFC-T165A mutant. The fusion of comFC with an αRep protein (named B2), for its structural study, is described in[35].

Expression of ComFC or its mutant forms in BL21(DE3) Gold strain was performed in 800 ml 2xYT o/n at 37 °C after induction with 0.5 mM IPTG (Sigma). Cells were harvested by centrifugation, resuspended in buffer 500 mM NaCl, 20 mM Tris-HCl (pH 7.5), 5% glycerol for ComFC-6His constructs, or in buffer 1 M NaCl, 100 mM Tris-HCl (pH 8), 100 μM TCEP for B2-ComFC-6His (SeMet labelled according to the protocol described in[56] and stored at −20 °C). Cell lysis

was completed by sonication (probe-tip sonicator Branson). After centrifugation at 20,000 g and 8 °C for 30 min, the proteins were purified on a Ni-NTA column (Qiagen Inc.), eluted with imidazole. ComFC-6His and B2-ComFC-6His were then desalted up to 100 mM NaCl and loaded respectively onto an Heparin and a MonoQ column (Amersham Pharmacia Biotech) and eluted with a gradient of NaCl (from 100 mM to 1 M). The proteins were desalted up to 200 mM NaCl and concentrated using Vivaspin 5,000 or 30,000 nominal molecular weight limit cut-off centrifugal concentrators (Sartorius), respectively, aliquoted, flash frozen in liquid nitrogen and stored at −80 °C, or dialysed in a 50% (vol/vol) glycerol buffer for storage at −20 °C.

SSB from *Deinococcus radiodurans* was expressed with a 6His-tag on N-terminal and purified as described before[57].

**Electrophoretic mobility shift assay.** DNA binding assays were performed by incubating indicated concentrations of proteins with fixed concentrations of Cy5-labelled DNA (Supplementary Table 3) in binding buffer (10 mM Tris-HCl pH 7.5, 50 mM KCl, 1 mM DTT, 0.1 µg/µl BSA) at 4 °C for 30 min. The nucleoprotein complexes were separated using native TBE-PAGE (6%). The gels were visualised by using a Typhoon imager. The depletion in substrate DNA was quantified using ImageJ by considering DNA without protein as 100%.

**Nuclease activity protection assay.** Fluorescent 78 bases long oligonucleotide Oso19 (5' Cy5-GCGTGGTGAAACCTACCTCTGATTCGAAACCTTCACTTT ACGTGGTCTGGCGTGGTGAATGTTCGTCGGCGTGCTCGA) (4 nM) was preincubated with or without drSSB or ComFC at the indicated concentrations in 10 µl of R buffer (25 mM Tris-HCl pH 7,5, 75 mM NaCl, and 0.1 mg/ml BSA) at 4 °C. Five units of Single strand endonuclease Mung Bean Nuclease (New England Biolabs) was added for 50 min at 37 °C. Reactions were stopped by addition of 150 mM EDTA and 0.5% SDS (final concentration) and incubated for 10 min with 1 µg of proteinase K (Thermofisher). Reaction products were then resolved by electrophoresis on a 10% native polyacrylamide gel in Tris-borate-EDTA (TBE 1X) buffer.

**Crystal structure determination.** SeMet modified B2-HpComFC-6His (12.5 mg/ml) was incubated with PRPP (3 mM) and MgCl2 (5 mM) at 4 °C. Crystals were grown in hanging drops by mixing the protein with reservoir solution in a 1:1 ratio. Crystals appeared after 5 days at 4 °C in 0.2 M Tri-potassium citrate + 18% PEG 3350. Glycerol cryo-protected crystals (two steps at 15 and 30%) were flash frozen in liquid nitrogen.

Diffraction data and refinement statistics are given in Table 1. Crystallographic data were collected at the selenium peak wavelength on the PROXIMA-2A from Synchrotron SOLEIL (Saint-Aubin, France) and processed with XDS[58] through XDSME (https://github.com/legrandp/xdsme). Four diffraction datasets collected from one crystal were merged to improve the experimental density map, by increasing the redundancy and the anomalous signal. Diffraction anisotropy was corrected using the STARANISO server (http://staraniso.globalphasing.org). The structure was solved by the SAD phasing method at 2.5 Å resolution using SHELX C/ D[59] to locate the 12 heavy atom sites, PHASER[60] to determine the initial phases and PARROT[61] to improve the phases by density modification, through the CCP4 programme suite[62]. The construction of the model was initiated using Buccaneer[63] and refined with the BUSTER using TLS and NCS restraints[64]. The model was corrected and completed using COOT[65]. The presence of a Zn2+ ion in the B2-HpComF structure was demonstrated by an energy scan performed on the crystals at the beamline (energy peak at 9.664 keV). Exploration of the 3D structures was performed using the following tools: Dali server[66], I-TASSER[67] and SWISS-MODEL servers[68] and PyMOL Molecular Graphics System (http://www.pymol.org).

**Bacterial two-hybrid assays.** The Bacterial Two-Hybrid test was used to probe interactions between proteins[69]. The full-length ComFC encoding sequence was fused to T18, at the C-terminal and N-terminal ends, T18-ComFC (pUT18C vector) and ComFC-T18 (pUT18 vector), respectively. Plasmids encoding T25-ComFC and ComFC-T25 were constructed using the pKT25 and pKNT25 vectors.

Plasmids encoding T18 and T25 fusion proteins were co-transformed in *E. coli* strain BTH101 and transformants were selected in Luria-Bertani agar plates containing kanamycin and ampicillin at 30 °C. Colonies were then spotted on plates containing kanamycin, ampicillin, IPTG and X-gal, incubated at 30 °C and stored at RT to follow the appearance and evolution of the blue colour.

**Differential scanning fluorimetry/thermal shift assay.** Purified protein (10.5 µg) was incubated with different analytes in reaction buffer (20 mM Tris-Cl, pH 7.5, 200 mM NaCl, 5X Sypro Orange). The temperature of the reaction mixture was raised from 25 °C to 95 °C. Shift in the fluorescence due to binding of the Sypro-Orange dye as the hydrophobic patches of the protein were exposed due to denaturation of the protein was recorded. The fluorescence maxima observed was used to calculate the approximate melting temperature of the protein in native conditions and in presence of the analyte.

**Reporting summary.** Further information on research design is available in the Nature Research Reporting Summary linked to this article.

## Data availability

The atomic coordinates and structure factors of B2-HpComFC have been deposited at the Brookhaven Protein Data Bank under the accession number 7P0H.

Other data are available in the main text or the supplementary materials. Data used for graphs and tables are available as a Source Data file. All other data can be requested to the authors. Source data are provided with this paper.

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

## Acknowledgements

We thank the beamline staff for assistance and advice during data collections at Synchrotron SOLEIL (Saint-Aubin, France; beamline PROXIMA 2). We thank Christopher Corbinais and Mariano Prado-Acosta for the construction of the initial *comFC* mutants. This work has benefited from the I2BC Macromolecular interactions measurements and Crystallisation facilities. Financial support for this work was provided by the Indo-French Centre for Promotion of Advanced research (CEFIPRA) grant 5203-5 (J.P.R., P.P.D.), Agence Nationale de la Recherche grant ANR-19-CE12-0003-01 (J.P.R.), French Infrastructure for Integrated Structural Biology (FRISBI) grant ANR-10-INSB-05-01 (S.Q.C., R.G.), Région Ile de France grant DIM1Health (J.P.R., P.P.D.), Commissariat à l'Energie Atomique (J.P.R., A.M.D.G., X.V., J.D.), Centre National de la Recherche Scientifique (S.Q.C., S.M.), *Enhanced Eurotalents* fellowship programme (CEA/EU) (P.P.D.) and Collectivité Régionale de Martinique (L.C.).

## Author contributions

Conceptualisation: S.Q.C., J.P.R. and P.P.D. Methodology: S.Q.C., S.M., P.P.D., A.M.D.G., X.V. and J.P.R. Formal analysis: H.W., J.V., R.G., S.Q.C. and J.P.R. Investigation: P.P.D., L.C., S.M.K., A.M.D.G., S.M., J.D., X.V., S.Q.C. and P.L. Data curation: L.C., H.W. and P.L. Writing - Original Draft: P.P.D., S.Q.C. and J.P.R. Writing – Review and Editing: P.P.D., S.Q.C. and J.P.R. All authors read and approved the manuscript. Visualisation: P.P.D., S.M.K., A.M.D.G., S.M., H.W., S.Q.C. and J.P.R. Supervision: J.P.R. and S.Q.C. Project administration: J.P.R. Funding acquisition: J.P.R., S.Q.C. P.P.D. and L.C. contributed equally to this work.

## Competing interests

The authors declare no competing interests.
