## [Peer Review File · Nature Communications]

Reviewers' Comments:

Reviewer #1:

Remarks to the Author:

The study by Damke et al. attempts to further our understanding of ComF. This protein is universally conserved in naturally competent species. Previous studies have shown that ComF mutants accumulate DNA in the periplasm, which suggest that they cannot translocate DNA into the cytoplasm. Here, the authors provide additional supporting evidence for a role of ComF in translocation of transforming DNA from the periplasm into the cytoplasm. And they demonstrate that ComF localizes to the inner membrane, however, it is unclear if this localization is critical for its function. The authors also provide preliminary evidence that ComF may play a secondary role in protecting or priming internalized DNA for recombination. This latter role is supported by previous studies which show that ComF likely interacts with DprA. The authors also solve the crystal structure of ComF and generate mutations to the PRPP domain and an N-terminal zinc-finger domain. They find that these mutations diminish ComF function in natural transformation assays. But a mechanistic understanding of how ComF, and the domains of the protein, promote DNA uptake and recombination remains lacking. While this study takes an important and major step towards improving our understanding of how ComF contributes to natural transformation, the results presented are fairly preliminary. The most significant advance is the crystal structure of ComF, however, using this new structural model to perform structure-function analysis or inform the mechanism of ComF also remains quite preliminary. Overall, the results are presented clearly and the experiments are done rigorously (with some exceptions noted below). The text, however, suffers from many grammatical errors and typos that need to be corrected.

Major:

At present, the observation that ComF localizes to the membrane is interesting, but is preliminary and lacks any substantial evidence to indicate that this localization is critical for its function. The authors suggest that ComFA may promote anchoring of ComF to the membrane, however they also state that this is not a conserved factor in many competent organisms. ComEC on the other hand is universally conserved in competent species and localized to the inner membrane. Why was ComEC not tested as the potential factor that anchors ComF to the membrane? This could be most easily tested by assessing ComF localization in a ComEC deletion mutant. Does localization of ComF change when cells are actively taking up DNA?

Fig 4a – why was the Δ comF mutant complemented with WT comF excluded from this assay? This is an important control since complementation with mutant ComF alleles are tested in this assay. This WT copy of ComF should be expressed at the same ectopic location with the same promoter as the mutated alleles. Also, a bar to denote the limit of detection in the assay should be shown for the RecA mutant in this figure.

The observation that loss of ComF, but not ComEC, diminishes natural transformation when ssDNA is electroporated into cells is a compelling preliminary result to suggest that ComF participates in homologous recombination. Possibly via recruitment of other competence proteins or protection of ssDNA in the cytoplasm. But this should have been tested further in vitro using purified ComF. Does ComF binding protect ssDNA from DNases? Does ComF binding help evict SSB similar to DprA? Does ComF help load DprA onto ssDNA in vitro?

It is great that the authors have solved the crystal structure of ComF. But attempts to use the new structural model to help inform ComF function remain very preliminary. The authors have made mutations to the zinc finger and PRPP domains and shown that they diminish or eliminate ComF function, but a mechanistic understanding of how these mutations diminish ComF function remains lacking. For example, do mutations to the zinc finger domain or PRPP affect the membrane localization of ComF? This might help support functional relevance of the membrane localization observed in Fig 3. Do these mutations affect interactions with DprA? Can DNA still be internalized into the cytoplasm in the ComF T165A mutant or zinc finger mutant (as quantified in Fig. 2D)? The latter would be particularly compelling to test because it might help separate whether one of the mutated ComF alleles can still support translocation into the cytoplasm, but not homologous recombination.

Minor:

There are a large number of grammatical errors and typos throughout the entire text (too many to list) that should be corrected.

Line 80 – The mechanism underlying the initial capture of DNA has also been demonstrated in *Neisseria*, *Vibrio cholerae*, *Thermus thermophilus* and *Legionella*. This occurs through the activity of minor pilin(s). PMIDs: 23386723 and 29891864 and 31186316. bioRxiv preprint: <https://www.biorxiv.org/content/10.1101/2021.08.16.456509v1>

Line 83 – DNA uptake in many organisms is carried out by a bona fide type IV pilus (not a pseudo-pilus), which should be acknowledged = “type IV pilus or pseudopilus”. This has been formally demonstrated in *Vibrio cholerae*, *Streptococcus pneumoniae*, and *Neisseria* species, and may also be true in other species as they continue to be tested and studied.

Reviewer #2:

Remarks to the Author:

The authors report the structure of *Helicobacter pylori* ComF(C), involved in natural transformation in Gram-positive and Gram-negative bacterial species. To date, no structure for ComF(C) has been reported, and the authors show that it possesses phosphoribosyl transferase (PRT) and Zn-finger domains. They further show that ComF(C) is involved in DNA transport through the cell membrane and handling of ssDNA once delivered into the cytoplasm. While the findings are undoubtedly important, there are several weak points in the writing and particularly the structural analysis.

Page 4, line 92: the naming convention used is somewhat confusing. What (if any) is the difference between ComFC and ComF(C)?

Page 4, line 102: the name ComF may be misleading as it suggests ComFABC have been characterized together, rather than just the ComFC protein. I suggest using ComF(C) (or ComFC) for clarity.

Page 8, line 212: it would be helpful if the authors could provide more details about why crystallization attempts with the isolated protein failed, and what other approaches were tried prior to gene fusion with an artificial alphaRep binder.

Page 4, line 216: “cristallisation” should be corrected to “crystallization”.

Page 8, line 222: the quoted resolution limit is 2.56 Å, whereas the limit given in Table 1 is 2.5 Å. Please clarify. How was the resolution cut-off determined?

Four datasets collected together were merged, but this is only mentioned in a footnote in Table 1. More details should be provided in the Methods section, page 17.

Page 8, line 223: the alphaRep-ComFC fusion protein was crystallized with four copies in the asymmetric unit. A figure should be included showing the packing of the fusion proteins in the asymmetric unit.

Page 8, line 228: the presence of alphaRep prevented the authors from drawing any conclusions about quaternary structure from the crystal structure. The authors used bacterial two hybrid assays to show that ComFC can interact with itself, but the suggestion ComFC could form head to tail dimers is poorly supported by evidence. The authors should provide additional experimental evidence to support dimerization or remove this statement.

Page 10: the authors state “an electron density that can correspond to an Mg²⁺ ion is present close to the PRPP”. Insufficient details are given about the Mg ion. How is it coordinated? Does it interact with PRPP? Is it required for activity? Is the presence of Mg a common feature among

PRPP-binding domains? Figure 5C provides no useful information about the Mg ion and should be redrawn or an additional panel added.

Page 10, line 289: the authors state that this is the first structure of a ComFC protein, yet the PRPP-binding domain is present in a large variety of proteins. How similar is the ComFC PRPP-binding domain to other known structures?

Page 11, line 317: are there any structural clues that might suggest why ComFC PRTase domain can bind monophosphorylated nucleotides and nucleotide di-phosphates in addition to PRPP? In my opinion, greater effort could be made to compare ComFC with other PRTase domain structures.

Figure 5: what is the extent of the interactions between the ComFC and alphaRep proteins? From the view shown, it appears that the alphaRep protein interacts with the Zn-finger and the PRPP. This should be discussed in the manuscript. The view shown in Figure 5 is not terribly informative in this regard and an alternative view (or views) would be helpful.

Figure 6: the authors use AMP to confirm that ComFC binds to nucleotides through its PRTase motif, but crystallized ComFC with PRPP. Did they try to crystallize ComFC with AMP?

Figure 6C: does the presence of the alphaRep protein in the fusion protein affect ssDNA binding?

Table 1: average B factor values should be quoted for ligands, ions and water molecules.

Table 1: (see comment above) what criteria were used to determine the resolution cut-off for the data?

Supplementary Figure 4: panel (A) showing the Zn-finger would be better as part of Figure 5 in the main manuscript.

The PDB validation report also raises some concerns:

(1) The residue property plots indicate several regions with poor fit to the electron density, particularly near the N-terminus of each protein. Without access to the structure and with no representative electron density figures, it is impossible to assess the quality of the structure. Representative electron density figures would be helpful.

(2) The validation report shows outliers in the bond angles and torsions of the PRPP molecule (PA and PB groups) in each chain. The authors are strongly advised to show representative electron density for the bound PRPP molecule, as well as for the Mg ion and Zn finger.

We would like to start by thanking both reviewers for their constructive comments and suggestions. By addressing them through both, new experiments and changes in the text, we believe the present version of the manuscript is stronger and that the results presented within support unambiguously the conclusions.

Response to Reviewer 1:

At present, the observation that ComF localizes to the membrane is interesting, but is preliminary and lacks any substantial evidence to indicate that this localization is critical for its function. The authors suggest that ComFA may promote anchoring of ComF to the membrane, however they also state that this is not a conserved factor in many competent organisms. ComEC on the other hand is universally conserved in competent species and localized to the inner membrane. Why was ComEC not tested as the potential factor that anchors ComF to the membrane? This could be most easily tested by assessing ComF localization in a ComEC deletion mutant. Does localization of ComF change when cells are actively taking up DNA?

We thank this reviewer for proposing these experiments. Following their suggestion, we have now analysed the subcellular localisation of ComFC in a *comEC* deletion mutant. As shown in the new Fig. 3c and its associated Supplementary Figure 2a, the absence of ComEC strongly reduces the fraction of ComFC present in the membrane fraction, supporting the conclusion that ComFC association with the membrane is at least partially mediated by ComEC.

Unfortunately, it is not possible to analyse the association of ComFC with the membrane specifically during active transformation, since we have no way so far to control the competence state in *H. pylori*. While there are some reports showing that the level of competence can be modulated, it is generally admitted that competence is constitutive in this species.

*Fig 4a – why was the $\Delta comF$ mutant complemented with WT *comF* excluded from this assay? This is an important control since complementation with mutant *ComF* alleles are tested in this assay. This WT copy of *ComF* should be expressed at the same ectopic location with the same promoter as the mutated alleles. Also, a bar to denote the limit of detection in the assay should be shown for the *RecA* mutant in this figure.*

We thank this reviewer for pointing out this important omission. As shown in the revised Figure 4a, ectopic expression of the wild type ComFC from the *rdx* locus allowed the complementation of the transformation phenotype. We have now done a new set of experiments to assess the impact of the ComFC mutant forms by comparing the yield of recombinants in cells expressing them with that of complemented cells expressing the wild-type protein with an equivalent construct. The new results, displayed in the new Figure 4b, show that expression of the mutant proteins is able to complement the *comFC* deletion mutant phenotype, therefore changing our conclusion. Once again, thank you for suggesting this critical control. The limit of detection of the assay, as defined by the potential presence of a single colony in the *recA* strain, is represented by a dotted line in Figures 4a and 4b.

*The observation that loss of *ComF*, but not *ComEC*, diminishes natural transformation when ssDNA is electroporated into cells is a compelling preliminary result to suggest that *ComF* participates in homologous recombination. Possibly via recruitment of other competence proteins or protection of ssDNA in the cytoplasm. But this should have been tested further in vitro using purified *ComF*. Does*

ComF binding protect ssDNA from DNases? Does ComF binding help evict SSB similar to DprA? Does ComF help load DprA onto ssDNA in vitro?

Following this interesting suggestion, we now show through a new experiment displayed in Fig. 4c that, indeed, purified ComF(C) is able to shield ssDNA from nucleases. This observation allows to propose a protective role for ComF(C) on the transforming DNA once it is internalised into the cytoplasm.

We are planning to study the functional and physical interactions between ComFC, DprA and SSB, at the biochemical level but, considering the amount of work and the time required to obtain significant results, we believe that these aspects are beyond the scope of the present manuscript.

It is great that the authors have solved the crystal structure of ComF. But attempts to use the new structural model to help inform ComF function remain very preliminary. The authors have made mutations to the zinc finger and PRPP domains and shown that they diminish or eliminate ComF function, but a mechanistic understanding of how these mutations diminish ComF function remains lacking. For example, do mutations to the zinc finger domain or PRPP affect the membrane localization of ComF? This might help support functional relevance of the membrane localization observed in Fig 3. Do these mutations affect interactions with DprA? Can DNA still be internalized into the cytoplasm in the ComF T165A mutant or zinc finger mutant (as quantified in Fig. 2D)? The latter would be particularly compelling to test because it might help separate whether one of the mutated ComF alleles can still support translocation into the cytoplasm, but not homologous recombination.

Once again, we thank this reviewer for their suggestion to perform new and informative experiments. We have now analysed the impact of the mutations in the different domains on several phenotypes: membrane localisation, DNA foci persistence and, as explained in the previous response, recombination efficiency after electroporation.

As shown in the new Fig. 3d and in Supplementary Figure 2b, while the T165A substitution in the PRPP binding domain did not affect the association with the membrane, the mutation of the Zn-finger precludes the presence of the protein in the membrane fraction, indicating that only the latter domain is required for the localisation of ComFC to the membrane.

A different scenario is observed for the internalisation of the DNA as shown in the new Fig. 2e. The persistence of the periplasmic foci in cells expressing either of the two mutant forms suggests that both domains are required for an efficient transport of the tDNA through the inner membrane.

Minor:

There are a large number of grammatical errors and typos throughout the entire text (too many to list) that should be corrected.

We apologise for these errors. We have now tried to the best of our knowledge to correct them.

*Line 80 – The mechanism underlying the initial capture of DNA has also been demonstrated in *Neisseria*, *Vibrio cholerae*, *Thermus thermophilus* and *Legionella*. This occurs through the activity of minor pilin(s). PMIDs: 23386723 and 29891864 and 31186316. bioRxiv preprint: [hQps://www.biorxiv.org/content/10.1101/2021.08.16.456509v1](https://www.biorxiv.org/content/10.1101/2021.08.16.456509v1)*

Line 83 – DNA uptake in many organisms is carried out by a bona fide type IV pilus (not a pseudopilus), which should be acknowledged = “type IV pilus or pseudopilus”. This has been formally demonstrated in Vibrio cholerae, Streptococcus pneumoniae, and Neisseria species, and may also be true in other species as they continue to be tested and studied.

In the new version we have acknowledged this and incorporated the relevant references.

Response to Reviewer 2:

Page 4, line 92: the naming convention used is somewhat confusing. What (if any) is the difference between ComFC and ComF(C)?

Page 4, line 102: the name ComF may be misleading as it suggests ComFABC have been characterized together, rather than just the ComFC protein. I suggest using ComF(C) (or ComFC) for clarity.

We did hesitate on the naming of the protein. As mentioned in the introduction, *comFC* is the only gene from the *comF* operon, first described in *B. subtilis*, to be conserved in all competent bacteria. As a consequence, and specially in Gram-negative bacteria where no *comFA* or *comFB* gene have been found, the *comFC* orthologue was in most cases annotated as simply *comF*. However, to avoid confusion and following this reviewer’s suggestion, we have now renamed the protein ComFC throughout the manuscript.

Page 8, line 212: it would be helpful if the authors could provide more details about why crystallization attempts with the isolated protein failed, and what other approaches were tried prior to gene fusion with an artificial alphaRep binder.

The purified protein, despite a search for the best purification and storage buffer, is unstable when concentrated to more than 1 mg/ml, which results in its precipitation. This prevented us from reaching the concentration allowing the nucleation in the crystallization tests. We therefore turned to alphaRep artificial proteins as stabilization and crystallization helpers. A sentence has been added in the text.

Page 8, line 216: “crystallisation” should be corrected to “crystallization”

This correction has been done.

Page 8, line 222: the quoted resolution limit is 2.56 Å, whereas the limit given in Table 1 is 2.5 Å. Please clarify. How was the resolution cut-off determined?

The resolution limit is actually 2.5Å. The data set have been cut in three dimensions by STARANISO. We have made the correction in the text.

Four datasets collected together were merged, but this is only mentioned in a footnote in Table 1. More details should be provided in the Methods section, page 17.

Four diffraction datasets collected from one crystal have been merged to improve the experimental map and increase the redundancy and the anomalous signal. A sentence explaining this approach has been added in the material and method section.

Page 8, line 223: the alphaRep-ComFC fusion protein was crystallized with four copies in the asymmetric unit. A figure should be included showing the packing of the fusion proteins in the asymmetric unit.

We propose the new Supplementary Figure 5 to show the four copies of the asymmetric unit. We calculated the interaction surfaces between these 4 molecules with the PISA server. This allows to show that only the dimer is "real" and that it is not due to the crystal packing. A sentence explaining this has been added in the text.

Page 8, line 228: the presence of alphaRep prevented the authors from drawing any conclusions about quaternary structure from the crystal structure. The authors used bacterial two hybrid assays to show that ComFC can interact with itself, but the suggestion ComFC could form head to tail dimers is poorly supported by evidence. The authors should provide additional experimental evidence to support dimerization or remove this statement.

Following this reviewer's suggestion, we have now removed the statement about the formation of head to tail dimers, which indeed, would require a much more in-depth analysis.

Page 10: the authors state "an electron density that can correspond to an Mg²⁺ ion is present close to the PRPP". Insufficient details are given about the Mg ion. How is it coordinated? Does it interact with PRPP? Is it required for activity? Is the presence of Mg a common feature among PRPP-binding domains? Figure 5C provides no useful information about the Mg ion and should be redrawn or an additional panel added.

We added an extra panel in Figure 5 (panel e) to show a zoom on the PRPP + Mg and the ComFC AAs that are interacting with it, by putting dotted lines with the distances (on the model of the Zn finger), and by adding the density of an omit-map around the PRPP + Mg to validate that there is indeed a ligand + ion there and to show how it fits inside. We also redid figure S4b with ligplot to add the Mg in addition to the PRPP.

Page 11, line 317: are there any structural clues that might suggest why ComFC PRTase domain can bind monophosphorylated nucleotides and nucleotide di-phosphates in addition to PRPP? In my opinion, greater effort could be made to compare ComFC with other PRTase domain structures.

In order to show that the PRPP loop is highly conserved in the PRTase family and does not predict which substrate will be bound, we selected the closest structural homologues of ComFC and superimposed this PRPP loop in the presence of different ligands. An additional figure (Supplementary Figure 6b) has been added, as well as a sentence in the text to explain this.

Figure 5: what is the extent of the interactions between the ComFC and alphaRep proteins? From the view shown, it appears that the alphaRep protein interacts with the Zn-finger and the PRPP. This should be discussed in the manuscript. The view shown in Figure 5 is not terribly informative in this regard and an alternative view (or views) would be helpful.

Please see the response to *Page 8, line 223*

Figure 6: the authors use AMP to confirm that ComFC binds to nucleotides through its PRTase motif, but crystallized ComFC with PRPP. Did they try to crystallize ComFC with AMP?

We obtained a large number of ComFC crystals in the presence of other nucleotides. They are shown in the figure below. But we never obtained usable data sets because the diffraction did not exceed 8 Å resolution. We have added a short sentence in the Material and Methods section to explain that.

Figure 6C: does the presence of the alphaRep protein in the fusion protein affect ssDNA binding?

We tested the interaction of the fusion with single and double stranded DNA by SPR. We observed only a weak interaction. AlphaRep appears to prevent access to the DNA binding site. This reinforces the hypothesis that it is via the Zn domain that ComFC appears to interact with DNA.

Table 1: average B factor values should be quoted for ligands, ions and water molecules. Table 1: (see comment above) what criteria were used to determine the resolution cut-off for the data?

We have now calculated the B-factor and produced a new Table I. STARANISO was used to determine the resolution cut-off.

Supplementary Figure 4: panel (A) showing the Zn-finger would be better as part of Figure 5 in the main manuscript.

We moved this figure as a panel in Figure 5 and added the 2Fo-Fc density around the Zn finger.

The PDB validation report also raises some concerns:

The residue property plots indicate several regions with poor fit to the electron density, particularly near the N-terminus of each protein. Without access to the structure and with no representative electron density figures, it is impossible to assess the quality of the structure. Representative electron density figures would be helpful.

It is normal that some flexible regions (ends, loops) do not fit well in the density. We can send the pdb + mtz files to check.

(1) The validation report shows outliers in the bond angles and torsions of the PRPP molecule (PA and PB groups) in each chain. The authors are strongly advised to show representative electron density for the bound PRPP molecule, as well as for the Mg ion and Zn finger.

We have now added the electron density around PRPP and Zn in the figures.

Reviewers' Comments:

Reviewer #1:

Remarks to the Author:

The revised manuscript from Damke et al. represents an improvement of the initial submission. While the molecular mechanism of ComFC still remains unclear, the new data in this revised manuscript help link the two domains of ComFC studied to a role of in DNA uptake across the inner membrane. The most significant contributions from this manuscript remain the structural analysis of ComFC and the demonstration that this protein binds to ssDNA. Both of which will help inform future research into the molecular mechanism of ComFC during natural transformation. Some of the other conclusions made from newly added data, however, need to be qualified or removed as discussed further below.

It is unclear how western blot quantification was normalized: "The ratios ComFC/MotB were normalized based on the membrane enrichment as monitored by MotB presence." Does this mean that the reported ComFC values were first normalized to the MotB levels in Supplementary Fig 2? If so, this seems appropriate and necessary, and should also be mentioned in the Figure legend. The differences in association appear very subtle, and it is not clear if they are even statistically significant. While a t-test cannot be run since all data is normalized to the WT, a one sample t-test / Wilcoxon signed-rank test should be run to see if the other samples are significantly different from 100%. If differences are not statistically significant, the authors should not be concluding that the comEC deletion affects ComFC localization. The same is true for the effect of the ComFC cysteine mutant.

The new data in Fig. 4a-b and the data in Supplementary figure 3 are actually not in agreement as stated in the text. While Fig. 4a-b demonstrate that complementation with WT or mutant ComFC rescues the defect observed in the comFC mutant, Supplementary Figure 3 shows that the mutant ComFC variants do not rescue the defect observed in the comFC mutant. Also, complementation with WT ComF is missing in the Supplementary figure. This needs to be resolved experimentally or discussed further in the manuscript.

Reviewer #2:

Remarks to the Author:

The manuscript by Damke and colleagues has been significantly improved following revision and I am satisfied with the responses to my comments.

There are still a number of language problems throughout and the manuscript would benefit from a careful proof-reading.

Response to reviewer #1

The revised manuscript from Damke et al. represents an improvement of the initial submission. While the molecular mechanism of ComFC still remains unclear, the new data in this revised manuscript help link the two domains of ComFC studied to a role of in DNA uptake across the inner membrane. The most significant contributions from this manuscript remain the structural analysis of ComFC and the demonstration that this protein binds to ssDNA. Both of which will help inform future research into the molecular mechanism of ComFC during natural transformation. Some of the other conclusions made from newly added data, however, need to be qualified or removed as discussed further below.

We thank this reviewer for their thorough evaluation of our manuscript. As detailed below, we have considered their suggestions and amended the text and relevant figures.

It is unclear how western blot quantification was normalized: "The ratios ComFC/MotB were normalized based on the membrane enrichment as monitored by MotB presence." Does this mean that the reported ComFC values were first normalized to the MotB levels in Supplementary Fig 2? If so, this seems appropriate and necessary, and should also be mentioned in the Figure legend. The differences in association appear very subtle, and it is not clear if they are even statistically significant. While a t-test cannot be run since all data is normalized to the WT, a one sample t-test / Wilcoxon signed-rank test should be run to see if the other samples are significantly different from 100%. If differences are not statistically significant, the authors should not be concluding that the comEC deletion affects ComFC localization. The same is true for the effect of the ComFC cysteine mutant.

We have now mentioned in the figure legend the normalisation and statistical analysis used. Moreover, the figures were replaced by new ones showing the individual values for the quantification. As mentioned in the legend a one-sample t test was carried out to determine the statistical significance of the difference between the wild-type and the *comEC* mutant (Suppl. Figure 2a) and between the expressed wild-type protein and its mutants (Suppl. Figure 2b). As indicated in the figure legend, p values were less than 0.001 for the differences between the wild-type and *comEC* strains (Supplementary Figure 2a) and between the wild-type and the C15SC18S ComFC proteins (Supplementary Figure 2b). Therefore, the statistical analyses support the conclusions that ComFC association with the membrane is dependent on the presence of ComEC, on one hand, and that the Zn finger domain integrity is required for presence of ComFC in the membrane fraction.

The new data in Fig. 4a-b and the data in Supplementary figure 3 are actually not in agreement as stated in the text. While Fig. 4a-b demonstrate that complementation with WT or mutant ComFC rescues the defect observed in the comFC mutant, Supplementary Figure 3 shows that the mutant ComFC variants do not rescue the defect observed in the comFC mutant. Also, complementation with WT ComF is missing in the Supplementary figure. This needs to be resolved experimentally or discussed further in the manuscript.

We apologise for the failure to remove the Supplementary Figure 3. The results being identical to those with the 75-mer for the strains we have tested, we did not redo the whole set with the 139-mer. We have now removed the figure which did not add new information.

Response to reviewer #2

The manuscript by Damke and colleagues has been significantly improved following revision and I am satisfied with the responses to my comments.

There are still a number of language problems throughout and the manuscript would benefit from a careful proof-reading.

We thank this reviewer for their positive return. The new version has been carefully reread and corrections to the text have been made.